# Genetic Mapping of Flavonoid Grain Pigments in Durum Wheat

**DOI:** 10.3390/plants12081674

**Published:** 2023-04-17

**Authors:** Natalia Sgaramella, Domenica Nigro, Antonella Pasqualone, Massimo Antonio Signorile, Barbara Laddomada, Gabriella Sonnante, Emanuela Blanco, Rosanna Simeone, Antonio Blanco

**Affiliations:** 1Department of Soil, Plant and Food Sciences (DiSSPA), University of Bari Aldo Moro, Via Amendola 165/A, 70126 Bari, Italy; natalia.sgaramella@gmail.com (N.S.); domenica.nigro@uniba.it (D.N.); antonella.pasqualone@uniba.it (A.P.); massimoantonio.signorile@uniba.it (M.A.S.); 2Institute of Sciences of Food Production (ISPA), National Research Council (CNR), Via Prov.le Monteroni, 73100 Lecce, Italy; barbara.laddomada@ispa.cnr.it; 3Institute of Biosciences and Bioresources, National Research Council (CNR), Via Amendola 165/A, 70126 Bari, Italy; gabriella.sonnante@ibbr.cnr.it (G.S.); emanuela.blanco@ibbr.cnr.it (E.B.)

**Keywords:** purple grains, wheat, grain color, nutritional quality, anthocyanins, transcription factors, regulatory genes, candidate genes

## Abstract

Pigmented cereal grains with high levels of flavonoid compounds have attracted the attention of nutritional science backing the development of functional foods with claimed health benefits. In this study, we report results on the genetic factors controlling grain pigmentation in durum wheat using a segregant population of recombinant inbred lines (RILs) derived from a cross between an Ethiopian purple grain accession and an Italian amber grain cultivar. The RIL population was genotyped by the wheat 25K SNP array and phenotyped for total anthocyanin content (TAC), grain color, and the L*, a*, and b* color index of wholemeal flour, based on four field trials. The mapping population showed a wide variation for the five traits in the different environments, a significant genotype x environment interaction, and high heritability. A total of 5942 SNP markers were used for constructing the genetic linkage map, with an SNP density ranging from 1.4 to 2.9 markers/cM. Two quantitative trait loci (QTL) were identified for TAC mapping on chromosome arms 2AL and 7BS in the same genomic regions of two detected QTL for purple grain. The interaction between the two QTL was indicative of an inheritance pattern of two loci having complementary effects. Moreover, two QTL for red grain color were detected on chromosome arms 3AL and 3BL. The projection of the four QTL genomic regions on the durum wheat Svevo reference genome disclosed the occurrence of the candidate genes *Pp-A3*, *Pp-B1*, *R-A1*, and *R-B1* involved in flavonoid biosynthetic pathways and encoding of transcription factors bHLH (*Myc-1*) and MYB (*Mpc1*, *Myb10*), previously reported in common wheat. The present study provides a set of molecular markers associated with grain pigments useful for the selection of essential alleles for flavonoid synthesis in durum wheat breeding programs and enhancement of the health-promoting quality of derived foods.

## 1. Introduction

Wheat is one of the major cereal crops, grown on about 210 million hectares all over the world, and represents a primary source of carbohydrates and proteins for the human population. The major cultivated wheat species is common wheat (*Triticum aestivum* L. ssp. *aestivum*), whereas durum wheat (*T. turgidum* L. ssp. *durum*) accounts for about 6% of the total cultivated wheat area and 5% of total wheat grain production [1]. Durum wheat, more adapted to semi-arid climates than common wheat, is mainly grown in countries of the Mediterranean basin, in the Middle East, USA, Canada, Mexico, and some areas in Kazakhstan, Australia, India, and Argentina. Quality traits of durum grain such as protein content and gluten strength make durum wheat more suitable for preparing typical end products consumed worldwide, such as pasta, couscous, bulghur, and several types of leavened and flat breads [2].

Most of the modern durum cultivars are characterized by grains with yellow to amber color as a result of the intense breeding activity for this trait over the last decades to improve the yellow color of semolina and end products generally preferred by consumers. Several quantitative trait loci (QTL) and candidate genes for the carotenoid content were detected and successfully deployed in durum wheat breeding programs to improve the yellow color and the nutritional value of end products [3].

More recently, other wheat grain pigments, such as flavonoids (i.e., anthocyanins, proanthocyanidins and phlobafenes), have attracted the attention of nutritional science backing the development of functional foods with claimed health benefits [4]. Several lines of evidence suggest that a regular intake of flavonoid-rich food contributes to the reduction of overweight and obesity as well as several associated chronic diseases [5]. The functional properties of flavonoids are ascribed to their antioxidant activity, which is associated with some health benefits both for human and animal populations [6,7].

Among the natural sources of flavonoids, some wild wheats and tetraploid wheat landraces and varieties are characterized by red, brown, purple, black, and blue grains due to the accumulation of different flavonoid compounds in the bran and/or aleurone layer [8]. Several studies have been concerned with the development and production of purple wheat products, including bread, pancakes, biscuits, porridge, crackers, chapati, fresh and dried pasta, and beer, and their characterization and evaluation as functional foods [9,10,11,12,13,14].

Studies on wheat-colored grains started in the 19th century by using the red and purple characters as morphological markers to estimate the wheat cross-pollination rate, gamete transmission, production of hybrid seeds, as well as the identification of double haploids from anther culture, histological observation, and pigment localization [15]. The genetic control of purple grain color, determined by the presence of anthocyanins in the pericarp, was studied in common wheat, with reported results from one or two independent dominant genes to two complementary dominant loci for purple grain [8,16,17]. The use of molecular markers supported the genetic control of two dominant genes with complementary effects, *Pp-A3* and *Pp-B1*, located on chromosomes 2A and 7B, respectively [18]. In common wheat, subsequent studies confirmed *Pp-A3* mapping on the centromeric region of chromosome 2A and the presence of three homoeoloci, namely *Pp-A3*, *Pp-B3*, and *Pp-D1*, on the short arm of homoeologous chromosomes 7A, 7B, and 7D, respectively [19,20]. Red grain color, determined by the presence of proanthocyanidins and catechins in the aleurone layer of the grain, was found to be controlled in common wheat by homoeoloci *R-A1*, *R-B1*, and *R-D1*, located on the long arm of chromosomes 3A, 3B, and 3D, respectively [21]. It was found that the dominant alleles have additive effects, while a single locus is sufficient to confer the red color on the grains [22]. The dark blue color of the grains is determined by a high content of delphinidins in the aleurone layer [23]. The blue color is controlled by the dominant gene *Ba1*, transferred from chromosome 4el of *Thinopyrum ponticum* (Podp.) into the long arm of chromosome 4B [24]. A second gene for the blue grain (*Ba2*) was transferred to common wheat from *T. boeoticum* as a disomic substitution of 4A (4AmL) [25].

The purple grain color was primarily found in tetraploid wheat populations grown on limited areas in Ethiopia and then transferred to common wheat in early breeding programs [26]. Studies on the inheritance of grain pigments concerned F_3_ segregating progenies of common wheat, with phenotypic assessment of grain color visually scored; only a few studies focused on durum wheat [19,27]. Therefore, the aim of this work is to study the genetic control of grain pigmentation in durum wheat. For this purpose, a segregant population of recombinant inbred lines (RIL), derived from crossing an Ethiopian purple accession of durum wheat and a modern durum variety, was used. The mapping population was genotyped with SNP markers and evaluated in four environments for total anthocyanin content (TAC), grain color, and the L*, a*, b* color index of whole meal flour. Moreover, a candidate gene analysis was conducted to uncover putative genes involved in the regulation of flavonoid biosynthesis. The reported results and the identified markers linked to TAC and grain color could support marker-assisted selection (MAS) in durum wheat breeding programs aimed to improve the quality of durum wheat grains and derived foods and to improve sustainable food and nutritional security.

## 2. Results

### 2.1. Phenotypic Variations of Total Anthocyanin Content (TAC) and Grain Color

The durum wheat RIL population obtained by crossing PG2 x Grecale and the two parental lines were evaluated for TAC and grain color in four replicated field trials carried out at Valenzano (Italy) in 2020, 2021, and 2022 and at Bari (Italy) in 2021. Highly significant differences (*p* < 0.01) among RIL genotypes in each of the two replicated trials were detected by analysis of variance for TAC (Appendix A); the combined analysis across environments revealed significant differences for years, genotypes, and environment x genotypes interaction (Appendix A). Table 1 reports the mean of the parental lines and the mean, standard error, range, genetic variance, and broad-sense heritability of the RILs in each environment. The parental lines were always significantly different for TAC in each environment; the Ethiopian line PG2 always had higher values (from 25.83 to 52.05 µg/g) than those of Grecale (from 1.41 to 4.29 µg/g). High broad-sense heritability values were found in the two replicated trials (0.81 in VAL_2022 and 0.91 in VAL_2021) and across environments (0.94). The RIL population showed mean values (8.06 to 17.91 µg/g) that were nearer to the means of Grecale than to those of PG2, while transgressive segregation was observed at high values falling outside the range of the parental lines in all environments. The frequency distributions (Figure 1 and Appendix A) were shifted toward low TAC values, and significant deviations from normal distribution were detected for the four trials and across environments. The normality of the original TAC values was not improved by the square root, arcsine, and log transformation (data not shown). Significant correlations for TAC were observed among the four field trials (Appendix A), with correlation coefficients ranging from r = 0.90 ** to r = 0.95 **.

Grain color was visually determined by using a 0–3 scale based on the intensity of the color (0 = amber, 1 = red, 2 = red-brownish, 3 = dark purple) (Figure 2), with the parental lines PG2 and Grecale always showing 0 and 3 scores, respectively. The grain color ranged from 0 to 3 in the whole segregant population, with values shifted toward low scores and with a high frequency of score 1 (from n = 64 to n = 88 in the four trials) (Figure 3). The color scores were highly consistent among the four experiments (Appendix A) with correlation coefficients, ranging from r = 0.82 ** to r = 0.88 **, indicating a relatively low genotype x environment interaction (Appendix A).

Color values L*, a*, and b* were also measured in each trial to investigate their possible relationships with TAC (Table 1). The parental lines were always different for each color value, while the RIL population showed mid-parental means. High broad-heritability values were observed for each color index. Correlation coefficients among all five traits (TAC, grain color, and L*, a*, and b*) in each environment are reported in Appendix A; correlation coefficients across environments are shown in Table 2. Across environments, L* and b* were found significantly negatively correlated with TAC and grain color, while a* showed positive correlation with TAC and grain color. TAC and grain color demonstrated a significant positive correlation (r = 0.63 **).

### 2.2. Genetic Linkage Map

The RIL population and the parental lines were genotyped by the wheat 25K SNP Illumina Infinium Array [28], including 24,145 single nucleotide polymorphisms (SNPs). After SNP data quality evaluation and control, a total of 5942 (24.6%) polymorphic markers were used to obtain a genetic linkage map via the software IciMapping v. 4.2 [29]. The ultimate map comprised 11 LGs assigned to individual chromosomes and 6 LGs assigned to 3 chromosomes (2B_1, 2B_2, 3A_1, 3A_2, 5B_1, 5B_2) (Appendix A). The total length of the A genome chromosomes was 1380.9 cM and included 2600 markers, while the total length of the B genome chromosomes was 1309.5 cM and included 3234 markers. Individual chromosomes had a length ranging from 166.1 cM (4B) to 240.2 cM (7A), and a number of markers from 265 (4B) to 604 (2B). The SNP density varied from a minimum of 1.4 markers/cM for chromosome 4A and a maximum of 2.9 markers/cM for chromosomes 5B and 6B.

### 2.3. Detection of QTL for Total Anthocyanin Content (TAC)

QTL for TAC were determined via the Inclusive Composite Interval Mapping (ICIM) method [29], and QTL detected in at least two environments with LOD values >3.0 were considered. Putative QTL for TAC and their main characteristics (LOD score, genetic and physical map position, closest markers, phenotypic variance explained, additive effects) in individual environments and across environments are reported in Table 3 and Figure 4. Two QTL (*QTAC.mgb-2A* and *QTAC.mgb-7B*) localized on chromosome arms 2AL and 7BS were identified via the ICIM-ADD method. These QTL were consistent in all four environments, and each explained 8.8–13.4% (QTAC.mgb-2A) and 11.3–16.9% (*QTAC.mgb-7B*) of TAC variation. Both QTL were also significant across environments with a phenotypic variance explained (PVE) of 10.5% and 16.7%, respectively. The positive alleles (elevated TAC) were both contributed by Ethiopian line PG2, with allelic effects ranging from 2.9 µg/g (*QTAC.mgb-2A* at VAL_2020) to 11.2 TAC µg/g (*QTAC.mgb-7B* at VAL_2021) in the different environments.

The ICIM-ADD method used for biparental populations can reliably detect QTL with additive effects [31], while the detection of epistasis effects is hampered by the complex pattern of epistasis, the large number of model effects, and the lack of efficient statistical methods [32,33]. Considering these issues and knowledge of the inheritance of purple grain color in common wheat and other cereal species [20,34], the interaction between the two detected QTL was analyzed by comparing the TAC means of the four groups of RILs with a different genotype at the two TAC loci. The genotype assignment to each RIL was carried out based on the closest SNP marker to each QTL. Mean values of TAC of the four groups of RILs with different genotypes at *QTAC.mgb-2A* and *QTAC.mgb-7B* in each of the four environments and across environments are reported in Figure 5.

By comparing the four RIL groups using Fisher’s LSD test, highly significant differences at *p* < 0.01 were found between the group with the dominant loci *QTAC.mgb-2A* and *QTAC.mgb-7B* and the other three groups having one or two recessive loci in individual environments and across environments, whereas non-significant differences were detected among the last three groups. The QTL analysis and the interaction between the detected loci are indicative of an inheritance pattern of two loci with complementary effect on TAC. The two QTL most likely correspond to the *Pp3* and *Pp1* genes for purple pericarp identified in common wheat [20], and, according to the catalogue of gene symbols for wheat [21], hereafter will be designated *Pp-A3* and *Pp-B1*.

### 2.4. Detection of QTL for Grain Color

Three QTL for grain color were detected on chromosome arms 2AL, 7BS, and 3BL. The first two QTL, *QGc.mgb-2A* and *QGc.mgb-7B*, significant in environments two and four, respectively, and with PVE ranging from 13.6% to 28.7%, were in the same genetic regions of the two detected loci for TAC (*QTAC.mgb-2A* and *QTAC.mgb-7B*) and should correspond to genes *Pp-A3* and *Pp-B1* identified in common wheat (Table 3). This validated the indication that the purple grain color can be attributed to TAC, as also indicated by the significant positive correlation between color intensity and TAC (r = 0.70 ***) (Table 2). The third QTL for grain color (*QGc.mgb-3B*), located on chromosome arm 3BL, was consistent in the four environments and across environments, and explained a phenotypic variance of 6.8–14.7%. This QTL is responsible for the red color categorized 1 and 2 in the current manuscript and should correspond to the locus *R-B1* identified in common wheat as responsible for the red grain color determined by proanthocyanidin and catechin compounds in the testa of seeds [22]. 

Considering the relatively low content of proanthocyanidin and catechin compounds determining the red color and the high content of TAC determining the dark purple color [35], and that this pigment could affect and partially mask the red color, a second QTL analysis was performed on a RIL subset excluding the homozygous lines for *Pp-A3/Pp-B1*-containing anthocyanin compounds. This further analysis on a subset of 110 RILs confirmed the presence of the red color locus *R-B1* on 3BL and detected a second QTL for the red grain color (*R-A1*) on chromosome arm 3AL. The two QTL were significant in all four environments and across environments, with LOD ranging from 4.6 to 8.6 and PVE from 14.8% to 23.1%. The statistical analysis of the four RIL groups differing in their *R-1* genotype showed significant differences in the double recessive genotype (*r-A1/r-B1*) compared to the other three genotypes (*R-A1/r-B1*, *r-A1/R-B1*, *R-A1/R-B1*) for grain color and for L*, a*, and b* color values, while no significant difference was found between the three genotypes having one or both dominant R-1 alleles (Table 4).

### 2.5. Candidate Genes

The 100 bp sequences of the closest SNP marker to each detected QTL on chromosomes 2A, 3A, 3B, and 7B (Table 3) were used to identify the corresponding genomic physical regions via a BLAST analysis against the durum wheat Svevo reference genome [30] (Table 3). The four identified QTL regions were further investigated within a linkage disequilibrium (LD) window of <5 cM to uncover candidate genes involved in the flavonoid synthetic pathways. Several dozen high-confidence protein-coding genes were found to be annotated, including the occurrence of previously reported candidate genes for purple and red grain color in common wheat. Table 5 reports the detected QTL for TAC and grain color and the corresponding durum wheat candidate genes along with the orthologous genes retrieved from the wild emmer wheat Zavitan reference genome [36] and from the common wheat Chinese Spring reference genome [37] for comparison. Phylogenetic studies were also performed to facilitate durum wheat candidate gene identification and classification. 

Within the chromosomal location of *QTAC.mgb-2A/QGc.mgb-2A*, a *Pp-A3* gene was identified, *TRITD2Av1G241410*, annotated as a member of the basic helix-loop-helix (bHLH) DNA-binding protein superfamily involved in flavonoid biosynthetic pathways [44]. A paralogous putative bHLH transcription factor (TF) gene, *TRITD2Av1G241610*, was also found within the same QTL interval. *TRITD2Av1G241410* was found to be homoeologous to *TRITD2Bv1G202430* located on the 2B chromosome, which is also a paralog of *TRITD2Bv1G202460*, another bHLH TF found in the same 2B region at about 98 kb. *TRITD2Av1G241610* had no homoeolog on the 2B chromosome. A phylogenetic tree (Figure 6 and Appendix A) was generated by processing these putative durum wheat bHLH TFs together with their corresponding orthologs from wild emmer and bread wheat bHLH TFs [47], along with the recently identified *TaMYC* TFs [38,41], and all the annotated Arabidopsis [48]. The phylogenetic analysis placed the four putative durum wheat bHLH TFs identified here into the cluster containing the majority of MYC TFs; in particular, they clustered with *TraesCS2A02G409400 (TaMYC1)*, [44] and with the Arabidopsis *AtbHLH42* [49].

A low-confidence gene, *TRITD7Bv1G027860*, was identified in the physical interval underlying the QTL *QTAC.mgb-7B/QGc.mgb-7B*. Interestingly, this gene showed a high sequence similarity to the durum wheat *TRITD7Av1G053110*, annotated as MYB transcription factor *TdMYB7A107* [42].

Candidate gene investigation was also performed for the physical regions of QTL for grain color detected on chromosomes 3A (*QGc.mgb-3A*) and 3B (*QGc.mgb-3B*). The *TRITD3Av1G260970* gene, also reported as *TdMYB3A056* and annotated as an MYB transcription factor [42], was found physically located within the chromosomal interval of *QGc.mgb-3A*. No orthologous gene was reported in common wheat, and no homoeologous gene on the B genome in durum wheat. A BLAST analysis allowed the identification of a high-similarity sequence, annotated as a low-confident gene on chromosome 3B, *TRITD3Bv1G255520*, also a MYB TF, which mapped within the *QGc.mgb-3B* interval. Taking advantage of a recent study on durum wheat MYB TFs [42], the putative durum wheat MYB TFs identified here were subjected to a phylogenetic analysis, including the annotated Arabidopsis *AtMYBs*, the emmer orthologs, and a subset of bread wheat MYB genes putatively involved in the regulation of phenylpropanoid biosynthesis. The durum wheat candidate genes and their wild emmer and bread wheat orthologs clustered with *AtMYB* TFs participating in the regulation of phenylpropanoid production [50] (Figure 6 and Appendix A).

## 3. Discussion

In recent years, durum wheat breeding programs have been focusing on several aspects of grain quality, such as protein content, starch content and composition, and grain color, a trait related to the presence of natural pigments, particularly carotenoids and anthocyanins. Pigmented wheats with red, purple, blue, and black grains have attracted the attention of the food industry for their large content of flavonoid compounds and associated antioxidant activity [51]. In particular, colored whole grains are regarded as promising new functional foods due to their anthocyanin content located in outer layers (pericarp, aleurone, testa) of grains [52]. Numerous in vivo and in vitro studies have demonstrated the high antioxidative activity of anthocyanins [23,53] and their health effects, including cardio-protection, anti-diabetes, anti-obesity, anti-cancer, and anti-aging-affiliated pathology [51,52,54]. As a result, the utilization of pigmented wheat grains to produce more nutritious and healthy food products has been considered [51]. To achieve this aim, a number of studies developed methods to carefully select the most appropriate milling fractions to produce flours rich in anthocyanins and other bioactive compounds present in the outer layers of the kernel without lowering the quality of end products [55,56,57,58,59].

Genetic, physiological, and biochemical studies have also shown that flavonoid compounds are involved in several biological activities, including plant defense responses to abiotic and biotic stress conditions [60,61]. In particular, anthocyanin-rich genotypes maintain significantly higher dry matter production under salt stress conditions [62]. Moreover, it was found that colored wheat genotypes are able to activate biotic and abiotic stress-responsive genes in response to drought [63], and to increase resistance to seed dormancy and pre-harvest sprouting [39,64]. In fact, anthocyanin biosynthesis and accumulation are enhanced by a number of hormones and environmental factors, including light, UV irradiation, high temperature, and heavy metals, besides drought and salinity [65].

### 3.1. Trait Variation and Genotype x Environment Interaction

Total anthocyanin content and composition of pigmented wheat grains and end products vary greatly depending on cultivar genotypes, pedological and climatic factors, agrotechnical management, extraction method, and processing conditions [66,67,68]. In the current work, an Ethiopian purple durum wheat line was crossed with an Italian amber durum wheat to obtain the RIL mapping population PG2 x Grecale, evaluated for five traits in four environments. The RIL population showed a wide variation for TAC in the different environments (ranging from 1.50–58.39 µg/g at VAL_2020 to 2.08–198.56 µg/g at VAL_2021), while the Ethiopian purple durum line PG2 had an average TAC value ranging from 25.83 µg/g at VAL_2020 to 52.05 µg/g at VAL_2021, thus indicating the influence of environmental factors on phenotypic expression of TAC. Despite the significance of genotype x environment interaction, TAC heritability showed high values (0.81–0.91 in individual environments and 0.94 across environments measured on a mean basis). TAC was always positively correlated with grain color and a*, and negatively correlated with L* and b* in all environments. The grain color, detected via a 0–3 scale based on color intensity, showed high heritability values (0.70–0.96) and significant interactions with environmental factors, especially for the variation of classes 1 and 2 in the four environments (Figure 3).

### 3.2. QTL for TAC and Grain Color

Early studies on the genetics of purple grain color in common wheat reported various results by using different purple wheat accessions in crosses with white grains: one dominant gene for purple grain (ratio 3:1 in F_2_ populations) [69,70,71], two duplicate dominant genes (ratio 11:5) [72], two independent genes (15:1) [66,73], and two complementary dominant genes (ratio 9:7) [72,74]. The use of molecular markers supported the genetic control of two dominant genes with complementary effects, *Pp-A3* and *Pp-B1*, located on chromosomes 2A and 7B, respectively [18]. Subsequent studies confirmed the *Pp-A3* mapping on the centromeric region of chromosome 2A and the presence of three homoeoloci, *Pp-A3*, *Pp-B3*, *Pp-D1*, on the short arms of the homoeologous chromosomes 7A, 7B, and 7D, respectively [19,20].

Most of these investigations were carried out in common wheat by visually scoring the grain color on F_3_ segregating populations. In contrast, very few genetic studies have considered the pigmented durum wheat [19,27]. The present research was conducted on an RIL population of durum wheat genotyped by the SNP 25K chip array, and simultaneously assessing the TAC, grain color, and the L*, a*, and b* color values for their possible correlation. QTL analysis identified, and validated, in durum wheat the two *Pp-A3* and *Pp-A1* loci, located on the long arm of chromosome 2A close to the centromere and on the short arm of chromosome 7B, respectively. The subsequent genetic analysis of the interaction of the two loci showed the absence of anthocyanin compounds in the three groups of RILs with a homozygous recessive genotype or with one dominant allele (*pp-A3/pp-B1*, *Pp-A3/pp-B1*, *pp-A3/Pp-B1*) and a high TAC content in the group of RILs with both dominant alleles (*Pp-A3/Pp-B1*), thus demonstrating the complementary action of the two loci on the phenotypic expression of TAC. The wide variation of TAC in the *Pp-A3/Pp-B1* RILs can be likely attributed to the segregation of the structural genes coding for the flavonoid compounds.

The genetics of the red grain color has been investigated since the beginning of the last century, with the well-known study of Nilsson-Ehle [75] on common wheat reporting the genetic control by three loci with additive effects; the experiment is considered a classic case of genetic dissection of a quantitative trait. Subsequent cytogenetic analyses located the 3 loci, designated *R-A1*, *R-B1*, *R-D1*, on the homoeologous chromosomes 3A, 3B, and 3D, respectively, and showed that one functional dominant locus is sufficient for the expression of the red color [76,77,78]. A series of two-way crosses and the color categorization into six classes, based on color intensity after NaOH treatment of the seeds, showed that grain color is under maternal inheritance and that the red color intensity was related to the number of R-1 loci; the grain color reaction to NaOH of each R-1 gene was not determined [79]. The quantification of proanthocyanidins of single, double, and triple homozygous R-1 genotypes has recently showed that the proanthocyanidin level and the red color intensity are closely linked to the number of dominant *R-1* alleles [80]. Moreover, the significantly higher proanthocyanidin content detected in genotypes *R-B1* and *R-D1* compared to that of *R-A1* suggested that each locus could differently contribute to the flavonoid biosynthesis [80]. Our results validate earlier observations supporting that one functional dominant allele is sufficient for determining the red grain color [79] and that genotypes with two or more functional alleles could only slightly increase the color intensity [80]. However, the observed results could also be due to genotype–environment interactions and/or to errors of the visual categorization of the grain color (particularly the 1 and 2 scores).

Overall, the use of a high-density SNP map determined the precise genetic and physical localization of the *Pp3*, *Pp-B1*, and *R-1* loci, and identified several tightly associated SNP markers to be used in MAS in durum wheat breeding programs.

### 3.3. Candidate Genes and Transcription Factors

The focus of this study was to determine the genetic control of grain pigments in durum wheat. Using an RIL population evaluated in four environments, we identified two QTL for total anthocyanin compounds (*QTAC.mgb-2A* and *QTAC.mgb-7B*) and two QTL for purple grain color (*QGc.mgb-2A* and *QGc.mgb-7B*), mapping in the same genetic and physical regions of chromosomes 2A and 7B. Two other QTL for red grain color (*QGc.mgb-3A* and *QGc.mgb-3B*) were detected on chromosome arms 3AL and 3BL. The projection of these four QTL regions on the durum wheat Svevo reference genome [30] disclosed the occurrence of candidate genes for purple and red grain color underneath their physical interval. Within the chromosomal location of *QTAC.mgb-2A/QGc.mgb-2A* and *QTAC.mgb-7B/QGc.mgb-7B*, the genes *Pp-A3* and *Pp-B1*, respectively, were identified on the corresponding physical interval on the Svevo genome. These dominant genes with complementary effects were firstly reported as responsible for grain color in bread wheat [18]. Later studies [27] reported that the Pp genes acted as transcriptional factors of an anthocyanin synthesis network in the pericarp. More recent studies on the anthocyanin biosynthesis in common wheat by multi-omic approaches have established that the key loci Pp-A3 and *Pp-A1, Pp-B1,* and *Pp-D1* correspond to the regulatory genes encoding the transcription factors MYC (*Myc-A1*) and MYB (*Mpc1-A1*, *Mpc1-B1*, *Mpc1-D1*) localized on chromosomes 2A and 7A, 7B, and 7D, respectively [44,45,46,49,81].

Two paralogous genes were identified in durum wheat within the *QTAC.mgb-2A/QGc.mgb-2A* interval, *TRITD2Av1G241410* and *TRITD2Av1G241610*, annotated as a bHLH DNA-binding superfamily protein and bHLH transcription factor, respectively. The two sequences are orthologs to the bread wheat *TraesCS2A02G409400* and *TraesCS2A02G409600*, two tandem duplicated genes reported as *TaMYC1* (Chen et al., 2019) or *TaMYC3-A1* and *TaMYC3-A2* [41]. Two sequences were also identified on durum wheat chromosome 2B: *TRITD2Bv1G202430* (ortholog of *TraesCS2B02G428000*) and *TRITD2Bv1G202460*, two bHLH TFs which showed very high sequence similarity and gene structure conservation. No duplication was observed for the homoeologous genes in emmer and bread wheat. The existence of these close genes on chromosomes 2A and 2B could represent tandem duplications, as observed on 2A in bread wheat, and are considered the main cause of the increase of MYC members in wheat [41]. As reported by Bai et al. [41], the duplication of genes as well as of chromosomal segments represents the primary factors of genome evolution in plants. Phylogenetic analyses placed the durum wheat candidate TFs into a cluster, including four closely related Arabidopsis members (*AtbHLH42*, *AtbHLH12*, *AtbHLH001*, *AtbBHLH002*), which were found to physically interact with R2R3–MYB proteins, and all participating in the regulation of flavonoid pathways, namely proanthocyanidin and anthocyanin biosynthesis [82].

A durum wheat low-confidence gene for MYB TF, *TRITD7Bv1G027860*, was found in the physical interval of the projected *QTAC.mgb-7B/QGc.mgb-7B* QTL, while on the A genome, a high-similarity sequence was annotated as *TRITD7Av1G053110*, an MYB transcription factor reported as *TdMYB7A107* [42]. The missing annotation of the homoeologous gene on the durum wheat 7B chromosome might be depending on mutations and/or polymorphism at the *Pp-B1* locus. In fact, we found a truncated R2R3 domain at the N terminus of the Svevo sequence. A recent paper [40] reported several sequence variations of *Pp3 (TaPpm1)* and *Pp1 (TaPpb1)* loci in different colored wheat lines, identifying four allelic variants within the coding region of *Pp3* and two allelic variants in the promoter region for *Pp1*. Alignments of the orthologous genes *TraesCS7B02G070400,* TRIDC7BG010500, and *TRITD7Bv1G027860* with the four *Pp3* allelic variants showed that common wheat, wild emmer, and durum wheat *Pp3* genes were different from the four variants identified in Jiang et al. [40].

Candidate gene investigation performed for the physical regions of QTL for grain color on chromosomes 3A (*QGc.mgb-3A*) and 3B (*QGc.mgb-3B*) uncovered MYB-like transcription factors (TFs) at first designated *Myb10* [22,43]. The gene *TRITD3Av1G260970* was found to be physically located within the chromosomal interval of *QGc.mgb-3A* and reported as an MYB transcription factor (*TdMYB3A056*) [42]. No orthologous gene was reported in bread wheat, and no homoeologous gene on the durum wheat B genome. However, a BLAST analysis identified a high-similarity sequence annotated as a low confident-gene, *TRITD3Bv1G255520*, also reported as an MYB TF, which mapped within the *QGc.mgb-3B* interval. Previously, in bread wheat, the R genes for the red grain color, located on the long arms of homoeologous chromosomes 3A, 3B, and 3D, were found to correspond to the three MYB-type transcription factors *Tamyb10-A1, Tamyb10-B1*, and *Tamyb10-D1*, respectively, and involved in the activation of the flavonoid biosynthetic pathway in developing grain [22,43]. *Tamyb10* genes encode R2R3-type MYB domain proteins, and sequence variations were found within *Tamyb10-A1* and *Tamyb10-B1* in recessive *R-A1* and *R-B1* lines [22]. A deletion of the first half of the R2-repeat of the MYB region in *Tamyb10-A1* in Chinese Spring caused a loss of function, which explained why the gene is not annotated in the bread wheat genome. A deletion of 19 bp was also found in the third exon of *Tamyb10-B1* genes of recessive *R-B1* bread wheat lines in comparison with the sequence of the red grain cv. Norin 61 [37]. In Chinese Spring, the deletion was located at 757,919,833 bp in the gene *TraesCS3B02G515900*. We found that the same 19 bp deletion occurred in Svevo (*TRITD3Bv1G255520,* position 3B: 769,813,185), and a shorter one in the wild emmer Zavitan (9 bp deletion in *TRIDC3BG075410*, position 3B: 771,961,919) (Appendix A). Phylogenetic analyses highlighted that the above durum candidate TFs and their wild emmer and bread wheat orthologs fall into the Arabidopsis functional group of flavonoid regulators *AtMYB75*, *AtMYB90*, *AtMYB113*, and *AtMYB114*, which are known to control anthocyanin biosynthesis [50]. In particular, they cluster with the Arabidopsis *AtMYB123*, ortholog of maize *ZmC1*, which controls the biosynthesis of proanthocyanidins in the seed coat of *A. thaliana* [49], thus providing clues as to their putative function.

Other studies have demonstrated that mutations in the promoters or coding sequences of bHLH or MYB proteins affected the ability of TFs to activate the downstream structural genes, thus being responsible for diverse color distributions or intensities in fruits and rice [82,83,84,85]. In fact, all these regulatory genes (bHLH and MYB) work together to activate the expression of the structural genes (*PAL*, *CHS*, *CHI*, *F3H*, *F3’H*, *DFR*, *F3′5′H*, and *FLS*) involved in the flavonoid biosynthesis and accumulation in vegetative tissues and seeds [46,65]. Further sequencing analysis would be useful to define eventual different genome variations in both colored and non-colored grains of different species.

### 3.4. Concluding Remarks

Cultivated cereal species include genotypes with pigmented grain (red, purple, blue, black) having high levels of flavonoid compounds, which possess beneficial effects on human and animal health. Colored grain can be consumed as whole grain, utilized to produce a variety of fortified functional foods, or for preparation of antioxidant dietary supplements. Using a biparental segregating population of durum wheat genotyped with the 25K wheat SNP arrays and evaluated for TAC and grain color, four QTL corresponding to genes *Pp-A3*, *Pp-B1*, *R-A1*, and *R-B1*, responsible for the purple and red grain and involved in flavonoid biosynthesis pathways, were identified. The four experimental trials revealed the effect of environmental factors on total grain anthocyanin content. The availability of the genome sequence in durum wheat allowed for the identification and precise localization of the candidate genes corresponding to transcription factors and a comparison of them with the orthologous genes in emmer and common wheat. The present study provides a set of molecular markers to select the essential alleles for flavonoid pigment synthesis in durum wheat breeding programs.

## 4. Materials and Methods

### 4.1. Plant Materials

A segregant population of 144 recombinant inbred lines (RILs) was developed by crossing the Ethiopian durum wheat line PG2 with purple grain with Grecale, an Italian durum cultivar with amber grain, and by advancing single F_2_ plants to the F_7_ generation via the single seed descent method. The PG2 line was obtained by genealogical selection carried out in the durum landrace accession CI 14629 collected at Shewa, Ethiopia, and kindly provided by the United States Department of Agriculture (USDA-ARS), USA. Grecale is an elite durum cultivar grown in Italy for its quality traits, including a relatively high grain protein content. The RIL population and the parental lines were grown in the experimental fields of the University of Bari, DISSPA Dept. (Bari, Italy), across four field trials carried out for three growing seasons (2020–2022) at Valenzano, Italy, (designated VAL_2020, VAL_2021, VAL_2022) and for one year (2021) at Bari (Italy) (designated BA_2021) under rainfed conditions. The experimental design for each trial was a randomized complete block with three replications. The parental lines PG2 and Grecale were repeated three times in each replication. The experimental unit consisted of a 1 m row, 30 cm apart, with 30 germinating seeds per plot. At maturity, the plots were hand-harvested, and the spikes shelled with a micro-thresher.

### 4.2. Phenotypic Analysis

Grain color intensity of each RIL and parental line was visually determined by using a 0-3 scale based on the deepness of the color: 0 = amber, 1 = red, 2 = red-brownish, 3 = dark purple (Figure 1). Grains were ground to wholemeal flour by means of a laboratory mill (Cyclotec Sample Mill, Tecator Foss, Hillerød, Denmark) equipped with a 1 mm sieve. Wholemeal samples were stored at 4 °C to minimize pigment degradation by oxidative enzymes for a maximum of 24 h before analysis. Colorimetric indexes were determined using the reflectance colorimeter Chroma Meter CR-300 (Minolta, Osaka, Japan) equipped with a pulsed xenon lamp. Color values L*, measuring brightness, a*, measuring red to green, and b*, measuring yellow to blue [86], were used in subsequent analysis. 

Total anthocyanin content (TAC) was determined according to Abdel-Aal and Hucl [87]. Briefly, 10 mL of 85:15 (*v*/*v*) methanol/1 M HCl was added to 1 g of each sample, and TACs were extracted by an orbital shaker at 500 rpm for 30 min; the supernatant was collected after centrifugation at 12,000× *g* for 5 min. The pellet was re-extracted under the same conditions. The two supernatants were combined, and the absorbance determined at 535 nm by a Cary 60 UV-Vis spectrophotometer (Agilent Technologies, Santa Clara, CA, USA). All determinations were done at least in triplicate, and analytical results expressed on a dry matter basis as µg/g of cyanidin 3-O-glucoside (Phytoplan, Heidelberg, Germany) used as standard in the previously set calibration curve. Each replication of all the RIL and parental lines were evaluated for TAC in the trials performed at Valenzano 2021 and Valenzano 2022, while a mixture of wholemeal of the three replications was assessed for the trials conducted at Valenzano 2020 and Bari 2021.

### 4.3. DNA Extraction and Molecular Marker Analysis

DNA of each RIL and parental lines PG2 and Grecale was extracted from fresh leaves by using the GeneElute Plant Genomic Miniprep Kit (Sigma, Waltham, MA, USA). After checking DNA concentration and quality by agarose gel-electrophoresis and NanoDrop2000 (Thermo Scientific™, Waltham, MA, USA), each DNA sample was diluted to 50 ng/µL and sent to TraitGenetics GmbH (Gatersleben, Germany) [88] for the genotyping assays with the wheat SNP 25K chip array developed by Illumina CSPro^R^ (San Diego, CA, USA), as described by [28].

### 4.4. Genetic Linkage Map Construction

The RIL population and the parental lines were genotyped by the wheat 25K SNP Illumina Infinium Array [28], including 24,145 single nucleotide polymorphisms (SNPs). A total of 2142 (9.1%) markers failed, 602 (2.4%) markers had more than 10% missing data, 13,141 (54.4%) markers were monomorphic, and 2268 (9.4%) markers showed distorted segregation (at *p* ≥ 0.001 value). The remaining 5942 (24.6%) polymorphic markers were used to obtain a genetic linkage map. Seven RILs with more than 20% missing data were also excluded. Linkage analysis between markers and linear order determination of loci were performed by the software QTL IciMapping v. 4.2 [29] using, initially, an LOD value of 6.0 for grouping the markers. SNP data from the reference consensus durum wheat map [89] were used as anchor loci as well as for assigning linkage groups to specific chromosomes. Linkage groups belonging to the same chromosome were then re-analyzed at LOD 3.0 for possible assignment to a single linkage group. A total of 108 markers remained unlinked or assembled in small LGs with any chromosomal anchor locus or assembled in an LG with the same map position, and therefore not further considered. Map distances were calculated by the Kosambi mapping function. Linkage groups were designated according to the wheat chromosome nomenclature; chromosome name is followed by a consecutive number when two or more linkage groups are present for a chromosome.

### 4.5. Candidate Genes and Transcription Factors

The QTL genomic regions identified on chromosomes 2A, 3A, 3B, and 7B were projected on the durum wheat Svevo reference genome [30,90]. The physical intervals were then investigated to validate the occurrence of purple and red grain color candidate genes encoding for bHLH and MYB TFs within these intervals. Furthermore, orthologous genes in durum wheat were detected by searching the wild emmer wheat Zavitan reference genome [36,90] and the bread wheat Chinese Spring reference genome v1.0 [37,91].

The candidate durum and wild emmer wheat genes were translated, and the first longest variant was considered for phylogenetic analyses. Multiple sequence alignments were performed by MAFFT v. 7 online service v. 7 [92] using MAFFT alignment with L-INS-i preset. Full-length amino acid sequences of putative bHLH genes were aligned with Arabidopsis *AtbHLH* proteins [48], bread wheat *TabHLH* [47], and *TaMYC* TFs [38,41]. The longest isoform of putative durum and wild emmer wheat MYB proteins were aligned with the annotated Arabidopsis *AtMYBs* retrieved from Plant TFDB and annotated according to Jiang and Rao [93], the durum wheat *TdMYB* TFs [42], and a subset of bread wheat MYB genes, putatively involved in the regulation of phenylpropanoid biosynthesis, selected from the phylogenetic analyses carried by Blanco and coworkers [42]. Phylogenetic studies based on maximum likelihood (ML) analyses were performed using W-IQ-TREE [94], with 1000 ultrafast bootstrap replicates. Phylogenetic trees were imported and annotated with iTOL v. 6.7 [95]. The trees were rooted with CDC5 sequences.

### 4.6. Statistical and QTL Analyses

Analysis of variance (ANOVA) for each trait for each field trial and for the combined data across environments was performed by the software CoStat v. 6.4 [96]. Data are reported as means ± standard error (es). Genetic variance, environmental variance, genotypic x environmental variance, and broad-sense heritability were obtained by using variance component estimates. Means of parental lines were compared by an independent two-sample Student’s *t*-test. Comparison of the four RIL groups with different genotypes was carried out by Fisher’s LSD test. Simple phenotypic correlation coefficients among L*, a*, and b* color values, TAC (Pearson correlation coefficients), and grain color (Sperman rank correlation coefficients) were calculated for each environment and across environments. QTL mapping was carried out via the Inclusive Composite Interval Mapping (ICIM) method using the software IciMapping v. 4.2 [29]. 

QTL analysis was conducted for the mean values of each phenotypic trait for each field trial and for the overall mean across the four trials. A QTL was considered significant when one or more markers were associated with the trait at a threshold *p* value of 0.001 (−log10(P) ≥ 3.0). QTL are reported when detected in at least two environments to reduce the detection of false-positive QTL. The proportion of phenotypic variance explained (PVE%) and the additive effect were estimated for each detected QTL. The International Rules of Genetic Nomenclature for wheat were followed for the QTL designation [21]. The software MapChart v. 2.2 [97] was used for the graphical representation of linkage groups and QTL.

## Figures and Tables

**Figure 1 plants-12-01674-f001:**
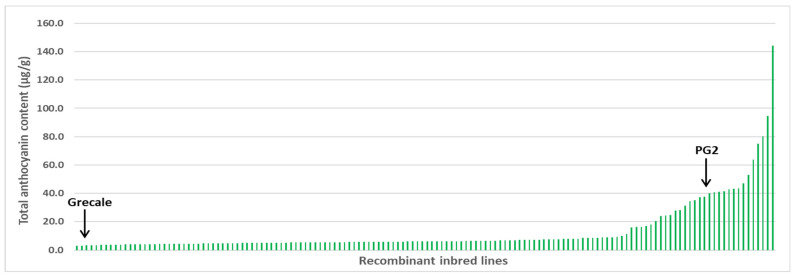
Total anthocyanin content in the PG2 x Grecale RIL population and parental lines. The reported values are the mean of four field trials carried out in southern Italy across 2020 and 2022.

**Figure 2 plants-12-01674-f002:**
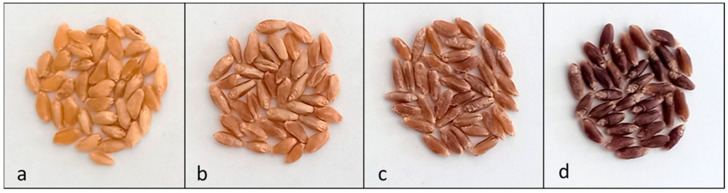
Seed color of test samples: (**a**) amber (score 0); (**b**) red (score 1); (**c**) red-brownish (score 2); (**d**) purple (score 3).

**Figure 3 plants-12-01674-f003:**
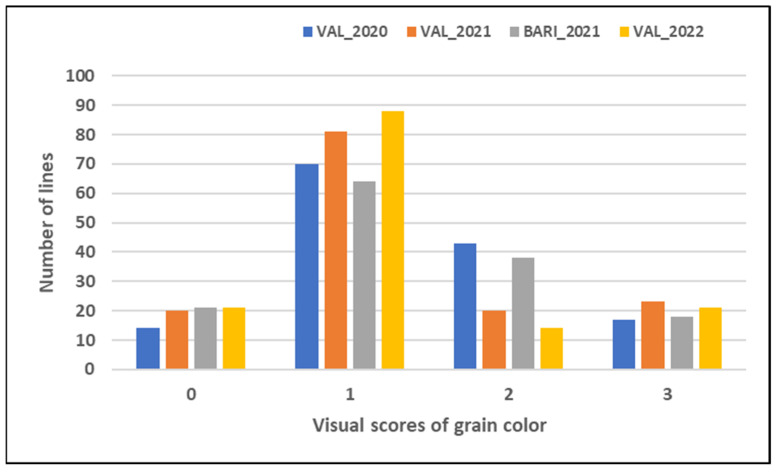
Frequency distribution of grain color scores evaluated using a 0 to 3 scale based on the deepness of the color (amber, light red, red-brownish and dark purple) in the biparental mapping population PG2 x Grecale grown in four environments (VAL_2020, VAL_2021, BA_2021, VAL_2022).

**Figure 4 plants-12-01674-f004:**
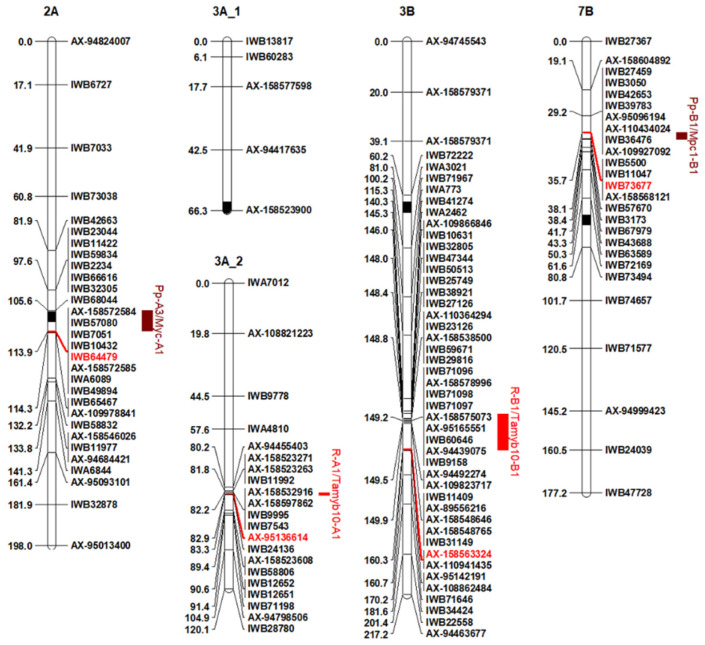
Schematic representation of four chromosomes of the durum PG2 x Grecale linkage map with positions of QTL for total anthocyanin content (TAC) and grain color. Each chromosome map is represented by the first and the last SNP marker, and by an SNP marker roughly every 20 cM. Markers are indicated on the right side and cM distances on the left side of the bar. QTL are represented by bars on the right of each chromosome bar. The closest marker to each QTL is indicated in red. The black segment on each chromosome bar represents the centromere.

**Figure 5 plants-12-01674-f005:**
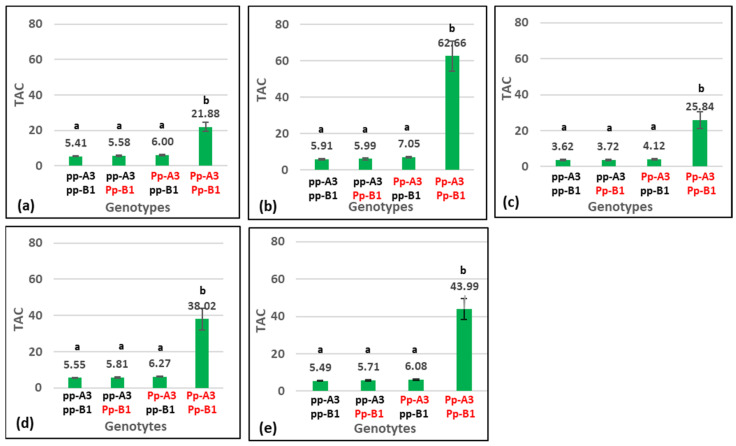
Mean value of total anthocyanin content (µg/g) (TAC) of groups of recombinant inbred lines with different genotypes at *Pp3* and *Pp1* loci grown in four environments: (**a**) Valenzano 2020; (**b**) Valenzano 2021; (**c**) Bari 2021; (**d**) Valenzano 2022; (**e**) across environments. *Pp-A3* and *Pp-B1* highlighted in red are dominant to *pp-A3* and *pp-B1*, respectively. Bars indicate mean value ± standard error. Different letters show significant differences (*p* < 0.05).

**Figure 6 plants-12-01674-f006:**
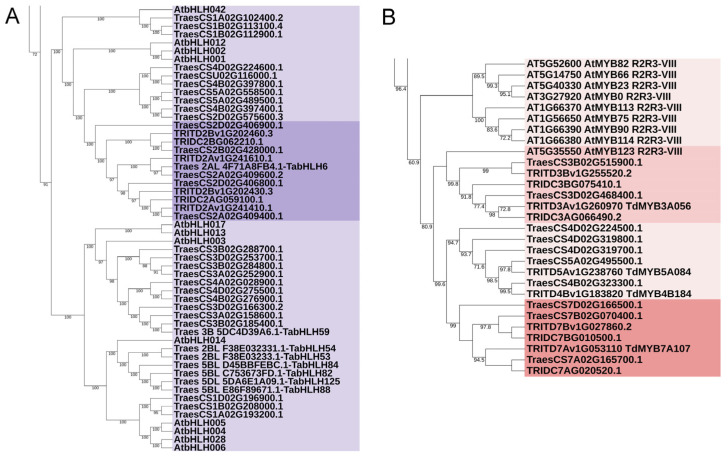
Phylogenetic relationships of durum wheat candidate transcription factors (TFs) with their homologs. (**A**) A section of the bHLH phylogenetic tree (complete tree in Appendix A), including the putative durum wheat TFs (TRITD) identified in this study, their wild emmer (TRIDC) and bread wheat (TraesCS) orthologs, and the Arabidopsis thaliana (AtbHLH) TFs. Purple: putative MYC TFs; dark purple: group with the Myc1 candidate genes identified here. (**B**) A portion of the R2R3-MYB phylogenetic tree (complete tree in Appendix A), including durum wheat (TRITD), their wild emmer (TRIDC) and bread wheat (TraesCS) ortholog TFs putatively involved in the regulation of phenylpropanoid biosynthesis, and A. thaliana (AtMYBs). Brown: putative MYB TFs; medium brown: group with the Myb10 candidate genes; dark brown: group with the Mpc1 candidate genes. Numbers on the branch nodes indicate bootstrap values (only bootstraps > 60 are shown).

**Table 1 plants-12-01674-t001:** Descriptive statistics and heritability of total anthocyanin content (µg/g), grain color (0–3 scores), and L*, a*, and b* color values of the recombinant inbred line population derived from the PG2 x Grecale cross grown across four environments.

Trait	Environment	Durum Wheat Parental Lines		RIL Population					
		PG2	Grecale	Mean	Standard Error	Minimum	Maximum	Variance	h^2^
Anthocyanin content (µg/g)	VAL_2020	25.83 **	3.92	8.86	0.69	1.50	58.39	68.71	°
	VAL_2021	52.05 **	4.29	17.91	2.37	2.08	198.56	808.78	0.91
	BA_2021	42.72 **	1.41	8.06	1.04	0.70	103.61	154.02	°
	VAL_2022	39.32 **	3.05	12.10	1.48	2.66	150.33	316.74	0.81
	Across environments	39.98 **	3.17	13.33	1.60	2.96	144.30	370.60	0.94 °°
Grain color (0–3)	VAL_2020	3 **	0	1.46	0.82	0.00	3.00	0.67	0.93
	VAL_2021	3 **	0	1.34	0.90	0.00	3.00	0.80	0.97
	BA_2021	3 **	0	1.36	0.87	0.00	3.00	0.75	0.95
	VAL_2022	3 **	0	1.26	0.84	0.00	3.00	0.71	0.95
	Across environments	3 **	0	1.37	0.82	0.00	3.00	0.68	0.97
L* color index	VAL_2020	73.83 **	81.9	78.6	1.8	73.7	83.1	3.2	0.88
	VAL_2021	75.4 **	84.0	80.7	2.2	74.3	84.7	4.7	0.97
	BA_2021	73.6 **	82.5	79.4	2.1	73.6	83.9	4.4	0.79
	VAL_2022	76.1 **	83.7	80.6	1.9	75.1	84.8	3.7	0.96
	Across environments	74.7 **	83.0	79.8	1.9	74.3	84.1	3.6	0.96
a* color index	VAL_2020	2.36 **	0.36	1.57	0.44	0.09	2.44	0.20	0.81
	VAL_2021	2.60 **	-0.11	1.38	0.58	−0.08	3.02	0.34	0.92
	BA_2021	2.64 **	0.62	1.65	0.50	0.43	2.91	0.25	0.86
	VAL_2022	2.38 **	0.02	1.36	0.50	0.04	2.71	0.25	0.91
	Across environments	2.49 **	0.22	1.49	0.47	0.19	2.55	0.22	0.94
b* color index	VAL_2020	11.7 **	16.8	14.6	1.3	10.6	18.2	1.7	0.92
	VAL_2021	12.1 **	18.2	15.3	1.8	8.9	18.9	3.1	0.95
	BA_2021	11.9 **	18.5	15.6	1.6	10.2	19.5	2.6	0.95
	VAL_2022	12.5 **	18.3	15.5	1.6	9.6	18.3	2.4	0.96
	Across environments	12.0 **	17.9	15.2	1.5	9.9	18.7	2.3	0.97

** Significant at *p* < 0.01 with a Student’s *t* test. ° h^2^ estimate is missing because TAC analysis was performed on mixed samples of three replications. °° h^2^, broad-sense heritability determined across two environments (VAL_2021 and VAL_2022).

**Table 2 plants-12-01674-t002:** Simple correlation coefficients among L*, a*, and b* color values, total anthocyanin content (TAC) (Pearson correlation coefficients), and grain color (Sperman rank correlation coefficients) from combined data of the recombinant inbred line population derived from the PG2 x Grecale cross grown in four environments.

Trait	L*	a*	b*	Grain Color	TAC
L*	1.00 **				
a*	−0.85 **	1.00 **			
b*	0.77 **	−0.76 **	1.00 **		
Grain color	−0.87 **	0.72 **	−0.72 **	1.00 **	
TAC	−0.73 **	0.47 **	−0.72 **	0.63 **	1.00 **

**: significant at 0.01 *p*.

**Table 3 plants-12-01674-t003:** Quantitative trait loci (QTL) for total anthocyanin content and grain color detected in the RIL population of durum wheat PG2 x Grecale via single-environment and cross-environment QTL analyses (ICIM-ADD).

QTL	Environment	Chrom. Arm	Genetic Position (cM)	Closest Marker	Physical Position (bp) ^a^	LOD	PVE (%)	Additive Effect
Total anthocyanin compounds								
*QTAC.mgb-2A*	VAL_2020	2AL	113	IWB64479	670,992,300	4.3	12.0	2.9
	VAL_2021	2AL	115	AX-109978841	672,230,615	4.7	13.4	9.9
	BA_2021	2AL	113	IWB64479	670,992,300	2.9	8.8	3.7
	VAL_2022	2AL	113	IWB64479	670,992,300	3.6	10.5	5.7
	Across environments	2AL	113	IWB64479	670,992,300	4.6	12.8	6.9
*QTAC.mgb-7B*	VAL_2020	7BS	39	IWB3173	89,972,586	5.0	14.4	3.2
	VAL_2021	7BS	38	IWB73677	83,850,422	5.9	16.9	11.2
	BA_2021	7BS	38	IWB73677	83,850,422	3.8	11.3	4.2
	VAL_2022	7BS	38	IWB73677	83,850,422	5.1	15.1	6.9
	Across environments	7BS	38	IWB73677	83,850,422	6.0	16.7	7.9
Grain color								
*QGc.mgb-2A*	BA_2021	2AL	113	IWB64479	670,992,300	2.9	6.3	0.2
	VAL_2021	2AL	118	AX-109978841	672,230,615	5.7	14.8	0.4
	VAL_2022	2AL	118	AX-109978841	672,230,615	5.5	13.6	0.3
	Across environments	2AL	118	AX-109978841	672,230,615	4.9	12.4	0.3
*QGc.mgb-7B*	VAL_2020	7BS	38	IWB73677	83,850,422	7.0	18.4	0.3
	VAL_2021	7BS	38	IWB73677	83,850,422	13.0	27.2	0.5
	BA_2021	7BS	38	IWB73677	83,850,422	10.4	23.4	0.4
	VAL_2022	7BS	38	IWB73677	83,850,422	13.1	28.7	0.5
	Across environments	7BS	38	IWB73677	83,850,422	12.4	27.3	0.4
*QGc.mgb-3B*	VAL_2020	3BL	161	AX-108862484	775,341,071	4.0	10.1	0.3
	VAL_2021	3BL	148	IWB47344	756,364,388	3.9	6.8	0.2
	BA_2021	3BL	161	AX-108862484	775,341,071	6.8	14.7	0.3
	VAL_2022	3BL	148	IWB47344	756,364,388	3.0	5.5	0.2
	Across environments	3BL	148	IWB47344	756,364,388	4.5	8.5	0.2
Grain color *								
*QGc.mgb-3A*	VAL_2020	3AL_2	83	AX-95136614	695,008,707	8.4	18.4	0.3
	VAL_2021	3AL_2	83	AX-95136614	695,008,707	4.6	14.2	0.2
	BA_2021	3AL_2	83	AX-95136614	695,008,707	8.6	23.1	0.3
	VAL_2022	3AL_2	83	AX-95136614	695,008,707	5.5	17.6	0.2
	Across environments	3AL_2	83	AX-95136614	695,008,707	7.8	20.5	0.2
*QGc.mgb-3B*	VAL_2020	3BL	158	AX-158563324	774,808,623	6.4	15.2	0.3
	VAL_2021	3BL	148	IWB47344	756,364,388	6.3	20.0	0.2
	BA_2021	3BL	148	IWB47344	756,364,388	8.2	21.8	0.3
	VAL_2022	3BL	148	IWB47344	756,364,388	5.1	16.2	0.2
	Across environments	3BL	148	IWB47344	756,364,388	8.1	21.3	0.2

^a^ The physical location is based on the durum wheat Svevo reference genome v1 [30]. * QTL analysis on a subset of 110 RILs excluding the lines with anthocyanin compounds. LOD, logarithm of odds; PVE, phenotype variance explained.

**Table 4 plants-12-01674-t004:** Mean values across four environments of grain color and L*, a*, and b* color values of groups of recombinant inbred lines with different genotypes at the *R-1* locus. Values followed by the same letter are not significantly different (*p* < 0.001).

Genotype	Grain Color	L*	a*	b*
*R-A1/R-B1*	1.4 b	79.8 b	1.6 b	15.41 b
*R-A1/r-B1*	1.2 b	80.0 b	1.5 b	15.63 b
*r-A1/R-B1*	1.2 b	80.1 b	1.5 b	15.55 b
*r-A1/r-B1*	0.3 a	82.4 a	0.8 a	16.71 a

**Table 5 plants-12-01674-t005:** Candidate genes for the detected QTL for total anthocyanin content and red grain color in durum wheat (*Triticum turgidum* ssp. *durum*) and related homoeologous genes in wild emmer wheat (*Triticum turgidum* ssp. *dicoccoides*) and common wheat (*Triticum aestivum*).

QTL	Species *	Ch.	Gene	*Triticum* ortholog TF **	ID	Position	Annotation
*QTAC.mgb-2A/QGC.mgb-2A*	*T. t. durum*	2A	*Pp-A3*	*Myc-A1.1*	*TRITD2Av1G241410*	661,787,635-661,794,834	bHLH DNA-binding superfamily protein.
	*T. t. durum*	2A	*Pp-A3*	*Myc-A1.2*	*TRITD2Av1G241610*	662,425,745-662,427,498	bHLH DNA-binding superfamily protein.
	*T. t. durum*	2B	*Pp-B3*	*Myc-B1.1*	*TRITD2Bv1G202430*	604,115,921-604,121,056	bHLH transcription factor.
	*T. t. durum*	2B	*Pp-B3*	*Myc-B1.2*	*TRITD2Bv1G202460*	604,213,485-604,217,695	bHLH transcription factor.
	*T. t. dicoccoides*	2A	*Pp-A3*	*Myc-A1*	*TRIDC2AG059100*	660,742,692-660,747,726	
	*T. t. dicoccoides*	2B	*Pp-B3*	*Myc-B1*	*TRIDC2BG062210*	611,414,598-611,419,239	
	*T. aestivum*	2A	*Pp-A3*	*Myc-A1.1*	*TraesCS2A02G409400*	667,010,726-667,015,774	MYC-like regulatory protein. *TaMYC1* [38]; *TaMYC3-A1* and *TaMYC3-A2* [39]; *TaPpb1* [40].
	*T. aestivum*	2A	*Pp-A3*	*Myc-A1.2*	*TraesCS2A02G409600*	667,647,089-667,652,609	MYC-like regulatory protein. *TaMYC1* [38]; *TaMYC3-A1* and *TaMYC3-A2* [39]; *TaPpm1* [40].
	*T. aestivum*	2B	*Pp-B3*	*Myc-B1*	*TraesCS2B02G428000*	615,352,721-615,356,928	*TaMYC3-B* [41]
	*T. aestivum*	2D	*Pp-D3*	*Myc-D1*	*TraesCS2D02G406900*	522,521,462-522,526,938	
	*T. t. durum*	7A	*Pp-A1*	*Mpc1-A1*	*TRITD7Av1G053110*	117,987,002-117,987,868	MYB transcription factor. *TdMYB7A107* [42].
*QTAC.mgb-7B/QTAC.mgb-7B*	*T. t. durum*	7B	*Pp-B1*	*Mpc1-B1*	*TRITD7Bv1G027860*	77,346,121-77,348,220	MYB transcription factor.
	*T. t. dicoccoides*	7A	*Pp-A1*	*Mpc1-A1*	*TRIDC7AG020520*	118,943,357-118,944,094	
	*T. t. dicoccoides*	7B	*Pp-B1*	*Mpc1-B1*	*TRIDC7BG010500*	81,313,184-81,314,083	
	*T. aestivum*	7A	*Pp-A1*	*Mpc1-A1*	*TraesCS7A02G165700*	121,397,856-121,399,015	
	*T. aestivum*	7B	*Pp-B1*	*Mpc1-B1*	*TraesCS7B02G070400*	77,430,167-77,432,792	*TaPpb1* [40].
	*T. aestivum*	7D	*Pp-D1*	*Mpc1-D1*	*TraesCS7D02G166500*	117242813-117245836	
*QGc.mgb-3A*	*T. t. durum*	3A	*R-A1*	*Myb10-A1*	*TRITD3Av1G260970*	694,983,038-694,991,683	MYB transcription factor. *TdMYB3A056* [42].
*QGc.mgb-3B*	*T. t. durum*	3B	*R-B1*	*Myb10-B1*	*TRITD3Bv1G255520*	769,811,153-769,814,329	MYB transcription factor.
	*T. t. dicoccoides*	3A	*R-A1*	*Myb10-A1*	*TRIDC3AG066490*	701,352,578-701,365,988	
	*T. t. dicoccoides*	3B	*R-B1*	*Myb10-B1*	*TRIDC3BG075410*	771,960,350-771,962,476	
	*T. aestivum*	3A			*-*		
	*T. aestivum*	3B	*R-B1*	*Myb10-B1*	*TraesCS3B02G515900*	757,918,264-757,920,082	*Tamyb10-B1* [22,43].
	*T. aestivum*	3D	*R-D1*	*Myb10-B1*	*TraesCS3D02G468400*	570,801,163-570,803,376	*Tamyb10-D1* [22,43].

* *T. t. durum* = *Triticum turgidum* ssp. *durum*; *T. t. dicoccoides* = *Triticum turgidum* ssp. *dicoccoides*. TF **, Transcription Factor *Myc-1* designation according to [44]; *Mpc-1* designation according to [45,46]; *Myb10* designation according to [22,43].

## Data Availability

All available data are contained within the article.

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
