# Peer review of "Genetic Mapping of Flavonoid Grain Pigments in Durum Wheat"

_plants, 2023, doi:10.3390/plants12081674_

Round 1

Reviewer 1 Report

1. The title is consistent with the study conducted.

2. The abstract is concise and relevant.

3. The material and study protocol is in accordance with the study carried out and well described.

4. The work has a medium degree of originality, with similar studies having been carried out previously.  The results obtained in the present study contribute to the field of durum wheat improvement through the set of molecular markers for the selection of essential alleles for the synthesis of the flavonoid pigment.

5. However, are needed minor corrections to improve the quality of the manuscript:

L66 – deleting author (Ficco et al., 2014)

However, are needed minor corrections to improve the quality of the manuscript:

L66 – deleting author (Ficco et al., 2014)

I suggest following replacements:

L560 – carried out  replaced with performed in Valenzano

L561 - evaluated replaced with assessed

Author Response

Reviewer #1 (R1)

Comments and Suggestions for Authors

  1. The title is consistent with the study conducted.
  2. The abstract is concise and relevant.
  3. The material and study protocol is in accordance with the study carried out and well described.
  4. The work has a medium degree of originality, with similar studies having been carried out previously. The results obtained in the present study contribute to the field of durum wheat improvement through the set of molecular markers for the selection of essential alleles for the synthesis of the flavonoid pigment.
  5. However, are needed minor corrections to improve the quality of the manuscript:

Authors (AA) - Thanks for your comments and suggestions. We have carefully considered and revised them accordingly. Please see response below.

R1 - L66 – deleting author (Ficco et al., 2014)

AA - (Ficco et al., 2014) has been deleted.

R1 - L560 – carried out replaced with performed in Valenzano

AA – “carried out” has been replaced with “performed”.

R1 - L561 - evaluated replaced with assessed

AA – “evaluated” has been replaced with “assessed”.

Reviewer 2 Report

The ms reports the genetic and molecular basis of pigmented grains in durum wheat. The size of the bi-parental population, genetic map density and number of environments for trait scoring are adequate for the conclusions made. The ms needs minor revision.

Line 121: here, h2 is 0.91 for TAC across environments, whereas in Table 1 the value for h2 is 0.94. Please also consider TAC h2 values presented in line 373. Table 1: Please indicate in the legend that h2 estimates are missing for TAC in Val_2020 and BA_2021 because of analysis of mixed samples. In line 562 you wrote Valenzano 2019, elsewhere you presented Valenzano 2020.

I recognize the introduction of gene designations according to the catalogue of gene symbols for wheat. Please be consistent with italics font for gene and QTL designations throughout the text and Table 5 as well as species names.

Lines 266-272: There is no need for a summary of this chapter. Please delete these lines because of redundant content

Although the order of sections agrees to the manuscript template, the position of the conclusion section is unlucky as it can be overlooked.

Author Response

Reviewer #2 (R2)

Comments and Suggestions for Authors

The ms reports the genetic and molecular basis of pigmented grains in durum wheat. The size of the bi-parental population, genetic map density and number of environments for trait scoring are adequate for the conclusions made. The ms needs minor revision.

Authors (AA) - Thanks for your comments and suggestions. We have carefully considered them and revised them accordingly. Please see response below.

R2 - Line 121: here, h2 is 0.91 for TAC across environments, whereas in Table 1 the value for h2 is 0.94. Please also consider TAC h2 values presented in line 373. Table 1: Please indicate in the legend that h2 estimates are missing for TAC in Val_2020 and BA_2021 because of analysis of mixed samples. In line 562 you wrote Valenzano 2019, elsewhere you presented Valenzano 2020.

AA – Done. We have checked the TAC h2 values and the correct value for TAC across environments is 0.94. Accordingly, 0.91 has been replaced by 0.94 in line 121.

In lines 372-374 the sentence has been modified: “…… TAC heritability showed high values (0.81-0.91 in individual environments and 0.94 across environments measured on mean basis).”

Table 1: the legend has been modified by adding “° h2 estimate is missing because TAC analysis was performed on mixed samples of three replications”.

Line 562: “Valenzano 2019” has been replaced by “Valenzano 2020”.

R2 - I recognize the introduction of gene designations according to the catalogue of gene symbols for wheat. Please be consistent with italics font for gene and QTL designations throughout the text and Table 5 as well as species names.

AA – Actually the submitted text had the italics font of genes, QTL and species names throughout the text and Tables (see the pdf format of submitted text). However, all revised text has been checked for the genes and species designation in italics font.

R2 - Lines 266-272: There is no need for a summary of this chapter. Please delete these lines because of redundant content.

AA – Accepted. The sentences have been deleted.

R2 - Although the order of sections agrees to the manuscript template, the position of the conclusion section is unlucky as it can be overlooked.

  1. We agree with your observation. The section “5. Conclusion” has been moved in the Discussion paragraph as “3.4. Concluding remarks”.

Reviewer 3 Report

Firstly, I would like to thank the authors for an excellent and robust study. I imagine the results will be of great use to the community for breeding higher anthocyanin content and it was very interesting to read about the complementary gene action of Pp-A3 and Pp-B1.

Some editing of typos and grammar is needed throughout, which I imagine the editorial team will take care of. Some examples of language that needs addressing:

Line 58: Should be "several lines of evidence". Phrasing

Line 64: Should be "and blue"

Line 65: Should be "aleurone layer"

Line 92: Phrasing

Line 106: Consider "to improve sustainable...." instead of "to gain..."

My comments regarding the rest of the manuscript and its scientific content are below:

Line 46: "Qualitative trait" - I think protein content and gluten strength can be classed as quantitative traits instead of qualitative traits?

Line 111: Consider moving sentence starting "TAC was evaluated on the whole meal flour..." to methods section.

Line 122: Should be "broad-sense heritability" not "broad-heritability". Please check other occurrences as well.

Table S1 and Table S2: Are there significance values that can be provided for the different components from these ANOVA tables (e.g. a p value for genotype, environment, and GxE)?

Table 1: Would be useful to mention that a t test is used to compare the parents. Ideally specify what type of t test was used.

Figure 1: Could be useful to include the parents in this figure to demonstrate transgressive segregation?

Table S3: I think it would be more appropriate to use Kendall's correlation/Spearman's rank correlation for Grain colour instead of Pearson's correlation as this data is ordinal.

Table 2: Same as above.

Line 170: I think the contents of section 2.2 might be better placed in section 4.4 as this concerns a lot of the same methodology.

Line 203: Are the TAC units micrograms per kg? If so, please state micrograms per kg instead of "units".

Line 229: Should be "p<" not "p>"?

Line 278: Is there a linkage disequilibrium decay plot that could be shown here to justify the <5cM window?

Lines 420 to 427 and Table 5 might be better placed within section 2.4.

Lines 507 to 508: Could these deletions in Zavitan and Svevo be illustrated with a figure in the results section or as a supplementary figure?

Line 573: Should this be "p < 0.001"?

Line 611: Please specify which type of t-test is used.

I hope these comments are useful and congratulations again on a great manuscript.

Author Response

Reviewer #3 (R3)

Comments and Suggestions for Authors

Firstly, I would like to thank the authors for an excellent and robust study. I imagine the results will be of great use to the community for breeding higher anthocyanin content and it was very interesting to read about the complementary gene action of Pp-A3 and Pp-B1.

Authors (AA) - Thanks for your comments and suggestions. We have carefully considered and revised them accordingly. Please see response below.

R3 - Some editing of typos and grammar is needed throughout, which I imagine the editorial team will take care of. Some examples of language that needs addressing:

- Line 58: Should be "several lines of evidence". Phrasing

- Line 64: Should be "and blue"

- Line 92: Phrasing

- Line 106: Consider "to improve sustainable...." instead of "to gain..."

AA - Actually, the submitted text had the italic font of genes, QTL and species names throughout the text and Tables (see the pdf format of the text). The revised text has been checked for the genes and species designation in italics font and for some editing of typos and grammar.

Your above indications have been accepted and the text has been corrected.

Line 92: The sentence has been phrased “The purple grain color was primarily found in tetraploid wheat populations grown on limited areas in Ethiopia and then transferred to common wheat in early breeding programs [26]. Studies on the inheritance of grain pigments concerned F3 segregating progenies of common wheat with phenotypic assessment of grain color visually scored; only a few studies focused on durum wheat [19, 27].”

R3 - My comments regarding the rest of the manuscript and its scientific content are below:

Line 46: "Qualitative trait" - I think protein content and gluten strength can be classed as quantitative traits instead of qualitative traits?

AA - In the context of the paragraph we were referring to the characteristics of the quality of the grain. Therefore "Qualitative trait" has been corrected in “Quality traits of durum grain…”

R3 - Line 111: Consider moving sentence starting "TAC was evaluated on the whole meal flour..." to methods section.

AA – Done. The sentence was deleted as already reported in the section 4.2. Phenotypic Analysis.

R3 - Line 122: Should be "broad-sense heritability" not "broad-heritability". Please check other occurrences as well.

AA – OK, "broad-sense heritability" has been used throughout the text.

R3 - Table S1 and Table S2: Are there significance values that can be provided for the different components from these ANOVA tables (e.g. a p value for genotype, environment, and GxE)?

AA - Table S1 and Table S2 have been integrated with F-value and Pr>F for genotype, environment, and GxE interaction.

R3 - Table 1: Would be useful to mention that a t test is used to compare the parents. Ideally specify what type of t test was used.

AA - The Student's t test used to compare the parents has been mentioned in Table 1.

R3 - Figure 1: Could be useful to include the parents in this figure to demonstrate transgressive segregation?

AA – Done. The mean value of parental lines has been indicated in Figure 1.

R3 - Table S3: I think it would be more appropriate to use Kendall's correlation/Spearman's rank correlation for Grain colour instead of Pearson's correlation as this data is ordinal.

R3 - Table 2: Same as above.

AA – Thanks for the statistic suggestion, the Spearman's rank correlation is more appropriate for Grain colour.  Therefore, Pearson's correlation for Grain colour has been replaced by the Sperman rank correlation coefficients in Table 2, Table S2 and Table S3, and indicated in the captions of the Tables.

R3 - Line 170: I think the contents of section 2.2 might be better placed in section 4.4 as this concerns a lot of the same methodology.

AA – Accepted. The sentences in lines 170-180 have been placed and adapted to section 4.4 Genetic Linkage Map Construction.

R3 - Line 203: Are the TAC units micrograms per kg? If so, please state micrograms per kg instead of "units".

AA - "units" has been replaced with µg/g.

R3 - Line 229: Should be "p<" not "p>"?

AA – Replaced.

R3 - Line 278: Is there a linkage disequilibrium decay plot that could be shown here to justify the <5cM window?

AA - We considered a linkage disequilibrium (LD) window of <5 cM between the values of 4.24, 5.0 and <10.0 cM determined in durum wheat genotypes by Maccaferri et al. (Genes 2022, 13(2), 293), Johnson et al. (Frontiers in genetics 2019, 10, 717) and Laidò et al. (PloS one 2014, 9(4), e95211), respectively.

R3 - Lines 420 to 427 and Table 5 might be better placed within section 2.4.

AA – Accepted. The sentences and Table 5 have been moved and adapted within section 2.4.

R3 - Lines 507 to 508: Could these deletions in Zavitan and Svevo be illustrated with a figure in the results section or as a supplementary figure?

AA –Done. Deletions are illustrated in Figure 4S and commented in the text.

R3 - Line 573: Should this be "p < 0.001"?

AA – OK, replaced.

R3 - Line 611: Please specify which type of t-test is used.

AA - Means of parental lines were compared by the independent two sample Student’s t-test. This was reported at line 611.